# The Development and Validation of an AI Diagnostic Model for Sacroiliitis: A Deep-Learning Approach

**DOI:** 10.3390/diagnostics13243643

**Published:** 2023-12-12

**Authors:** Kyu-Hong Lee, Ro-Woon Lee, Kyung-Hee Lee, Won Park, Seong-Ryul Kwon, Mie-Jin Lim

**Affiliations:** 1Department of Radiology, College of Medicine, Inha University, Incheon 22212, Republic of Korea; mdcappuccino@daum.net (K.-H.L.);; 2Division of Rheumatology, Department of Internal Medicine, College of Medicine, Inha University, Incheon 22212, Republic of Korea; parkwon@inha.ac.kr (W.P.);

**Keywords:** sacroiliitis, DEEP:PHI, deep-learning-based diagnostic tool

## Abstract

Purpose: Sacroiliitis refers to the inflammatory condition of the sacroiliac joints, frequently causing lower back pain. It is often associated with systemic conditions. However, its signs on radiographic images can be subtle, which may result in it being overlooked or underdiagnosed. This study aims to utilize artificial intelligence (AI) to create a diagnostic tool for more accurate sacroiliitis detection in radiological images, with the goal of optimizing treatment plans and improving patient outcomes. Materials and Method: The study included 492 patients who visited our hospital. Right sacroiliac joint films were independently evaluated by two musculoskeletal radiologists using the Modified New York criteria (Normal, Grades 1–4). A consensus reading resolved disagreements. The images were preprocessed with Z-score standardization and histogram equalization. The DenseNet121 algorithm, a convolutional neural network with 201 layers, was used for learning and classification. All steps were performed on the DEEP:PHI platform. Result: The AI model exhibited high accuracy across different grades: 94.53% (Grade 1), 95.83% (Grade 2), 98.44% (Grade 3), 96.88% (Grade 4), and 96.09% (Normal cases). Sensitivity peaked at Grade 3 and Normal cases (100%), while Grade 4 achieved perfect specificity (100%). PPVs ranged from 82.61% (Grade 1) to 100% (Grade 4), and NPVs peaked at 100% for Grade 3 and Normal cases. The F1 scores ranged from 64.41% (Grade 1) to 95.38% (Grade 3). Conclusions: The AI diagnostic model showcased a robust performance in detecting and grading sacroiliitis, reflecting its potential to enhance diagnostic accuracy in clinical settings. By facilitating earlier and more accurate diagnoses, this model could substantially impact treatment strategies and patient outcomes.

## 1. Introduction

Spondyloarthritis (SpA) consists of a group of interconnected rheumatic conditions that include ankylosing spondylitis, psoriatic arthritis, spondylitis associated with inflammatory bowel disease, and reactive arthritis [1]. And, it refers to a group of diseases marked by inflammation in the spine and peripheral joints, among other clinical manifestations. This spectrum includes axial spondyloarthritis (axSpA) and peripheral spondyloarthritis. Over the past few years, our understanding of SpA’s progression and underlying causes has deepened, paving the way for more effective treatments with an emphasis on early detection. For instance, Ankylosing spondylitis (AS), a type of axSpA, is a chronic inflammatory disease that primarily targets the spine and sacroiliac joints. It predominantly impacts the entheses and often leads to significant disability and a decreased quality of later life, especially since it tends to begin in early adulthood [2].

For many years, the detection of radiographic sacroiliitis has been the only way to confirm a definite diagnosis of the disease prior to the development of structural skeletal damage. Definite radiographic sacroiliitis, characterized by a minimum of grade 2 bilaterally or grade 3 unilaterally, is a required component of the modified New York criteria for diagnosing AS [3]. 

The sacroiliac joints are especially critical as they are commonly impaired. Computer tomography (CT) is frequently used to identify structural changes in the sacroiliac joints [4]. CT holds certain advantages over traditional radiography, such as its ability to provide multiplanar evaluations for enhanced analysis [5]. However, its usage is typically restricted due to concerns about higher radiation exposure compared to plain radiography [6]. Additionally, just like plain radiographs, CT is not effective in detecting sacroiliitis in its non-radiographic stages. And, one previous study reported the correlation between data from plain radiography and CT is relatively weak, with a kappa value of only 0.2418 [5]. Due to these limitations, one of the most important techniques for the diagnosis of sacroiliitis in diagnosing axSpA via magnetic resonance (MR) imaging.

For the early diagnosis of inflammation in the sacroiliac joints, MRI is considered the best method due to its superior ability to detect differences in tissue, highlighting conditions like subchondral bone marrow edema. The international group, Assessment of SpondyloArthritis International Society (ASAS), suggests using T2-weighted MRI sequences, specifically short tau inversion recovery (STIR), or T2 fat saturation (fat-sat) for identifying active inflammation in the sacroiliac joints. On an MRI, the signs of active inflammation manifest as bright intensity near the joint surface within the subchondral bone [7].

Magnetic resonance imaging (MRI) of the sacroiliac joints nowadays enables earlier diagnosis; definite radiographic sacroiliitis can be detected at the time of diagnosis in about 33% of patients with symptoms lasting up to 1 year and in about 50% of the patients with a symptom duration of 2 to 3 years [8]. Early and accurate detection is now crucial to provide timely and safe treatments [9]. Typically beginning in young adulthood, axSpA can lead to severe physical impairments and a diminished quality of life if not promptly diagnosed and treated. Given its early onset and the associated medical and treatment implications, axSpA has a notable socioeconomic impact [10]. However, one of the greatest challenges in managing as SpA still lies in its diagnosis. Its early stages can be difficult to discern through traditional diagnostic methods, leading to substantial delays in diagnosis and treatment initiation [11].

Radiographic imaging, particularly X-rays, plays a prevalent diagnostic tool in the diagnostic process eventually, although it comes with several limitations. Detecting subtle articular changes in the sacroiliac joints indicative of early axSpA can be challenging, and there is a degree of subjectivity and variability in interpreting these images rather than CT and MR scans [12]. Meanwhile, computer-aided diagnosis (CAD) now serves as a pivotal adjunct in clinical settings, offering a “second opinion” to medical professionals. While CAD systems have long supported clinicians, the incorporation of diverse machine-learning techniques has further enhanced their capabilities. Recent research indicates a rising trend in the efficacy of CAD systems, especially when they harness artificial intelligence (AI) methodologies [13].

The landscape of healthcare has been dramatically transformed by technology, particularly by the advent of artificial intelligence (AI). AI has permeated many aspects of healthcare, from risk stratification to diagnostic imaging, enabling more efficient and accurate diagnoses [14]. With the integration of AI, particularly machine-learning algorithms, we may overcome these hurdles and enhance the diagnostic accuracy of axSpA [15].

Deep learning, a subset in the field of AI, has achieved significant advancements in both kinds of medical data analysis nowadays. Notably, deep neural networks have shown a proficiency for identifying breast cancer from mammograms, categorizing skin cancer, and labelling chest X-rays [16,17,18]. Instead of building models from the ground up, these research endeavors employed transfer learning, leveraging pre-trained networks from unrelated domains. This method capitalizes on the knowledge of established models, cutting down the data required for training and enhancing the models’ accuracy. For instance, Ureten et al. effectively employed deep-learning networks combined with transfer-learning techniques on pelvic plain radiographs to diagnose hip osteoarthritis [19]. But, many researchers find it challenging to navigate complex coding systems and deep-learning platforms.

Our study aims to investigate the use of AI, based on a simple no-coding system platform, in identifying sacroiliitis associated with axSpA through X-rays. We hope to illuminate how the fusion of AI with diagnostic imaging could transform the approach to complex musculoskeletal disorders.

## 2. Materials and Methods

### 2.1. Patient Demographics

This study included a total of 492 patients who visited our hospital. Patients were randomly chosen from those who underwent SI joint CT scans (Revolution HD, General Electric Medical Systems, Waukesha, WI, USA) and plain radiographs at our institution between 2019 and 2022, without accounting for demographics like age and sex. The demographics of the patients are presented in Table 1.

### 2.2. Methodology

The patients were divided into two groups (refer to Table 1). The first group, termed the ‘non-sacroiliitis group’, comprised patients who did not exhibit sacroiliitis either radiologically (including X-ray and CT findings) or clinically. To constitute this group, two radiologists (R.W.L. and K.H.L.) with 9 years of experience in musculoskeletal radiology analyzed the patients’ images to confirm the absence of abnormalities in the sacroiliac joints. Additionally, the clinical diagnosis of non-sacroiliitis group was established based on the evaluations of two rheumatologists (S.R.K. and M.J.L.). This group included a total of 214 patients.

Furthermore, the second group included 278 patients with X-ray and CT-confirmed sacroiliitis. The sacroiliitis patients were further divided into four grades as follows: 54 patients were Grade 1, 101 were Grade 2, 78 were Grade 3, and 45 were Grade 4. All patients involved underwent an anteroposterior view of a routine pelvis X-ray. The standard reference for diagnosis relied on CT findings to establish the final grade of sacroiliitis. All the images were reviewed separately, and final consensus readings were made by radiology experts (R.W.L. and K.H.L.). If there was a discrepancy in the grading of sacroiliitis, they would view the images together and discuss to reach an agreement. To reduce discrepancies and confounding factors between imaging modalities, patients were included in the study only if CT findings were consistent with plain radiographic readings. X-rays demonstrated confounding factors such as degenerative changes were eliminated at that step. The sacroiliitis patients were further divided into four grades, using the New York Criteria (Table 2).

### 2.3. Dataset Proportion

The dataset, which included 492 patients from our study, was divided into a training set (*n* = 300, 61.0%), validation set (*n* = 96, 19.5%), and test set (*n* = 96, 19.5%). The sacroiliitis patients were further divided into four grades as follows, using the New York Criteria: 54 patients were grade 1, 101 were grade 2, 78 were grade 3, and 45 were grade 4 (Table 3). We employed deep-learning algorithms to train the data.

### 2.4. Image Preprocessing and AI-Learning Algorithms

For AI training purposes, only the image of the right SI joint was used. We then performed some preprocessing for training. The image acquired around the right SI joint was resized to 256 × 256 pixels, and the color information of the image was removed by color to grayscale processing. Z-score standardization was performed to adjust the dynamic range for consistency for subsequent image processing. This can be expected to improve computational efficiency. In addition, histogram equalization CLAHE processing was performed to improve the contrast of the image and adjust the histogram uniformly. This preprocessing was performed to make the characteristics of the image more prominent.

The Grad-CAM (Gradient-weighted Class Activation Mapping) technique was employed to investigate the rationale behind class predictions in a CNN-based deep-learning classification network [20]. By depicting the primary regions of the input image influencing the final outcome, Grad-CAM provides an intuitive validation of classification results, especially for plain radiographs indicating structural bone change.

The AI-learning model utilized was DensNET121 (epoch = 20, batch size = 4, learning rate decay = 1), and for the optimizer we used adam (learning rate = 0.0001, beta 1 = 0.9, beta 2 = 0.999, amsgrad = false). DensNet121 is a neural-network structure that applies the CNN deep-learning method with a dense connectivity method and has 201 convolutional layers. It is suitable for 2D classification, has efficient computation speed, and learns relatively well even with a small number of data sets, so we chose this method in this study because the number of data sets was small. The architecture of DenseNet-121 is illustrated in Figure 1.

All of this was performed on a no-code-based AI open platform (DEEP:PHI, ver.2.7.6; Deepnoid Inc. (Seoul, Republic of Korea), https://www.deepphi.ai, accessed on 31 August 2023) instead of with conventional coding tools. DEEP:PHI is a platform that facilitates image preprocessing and AI research using a GUI (graphic user interface), eliminating the need for coding. Being cloud-based, it allows for research even in the absence of substantial computing resources. The advantage of using this platform was that we could easily drag and drop AI training images using the GUI without coding ourselves (Figure 2).

## 3. Result

While performing 20 epochs, the processing time was 13.2243 s per epoch. In the early stages of training, the AI algorithm showed a rather low diagnostic accuracy, but as the epochs were repeated, the diagnostic performance improved (Figure 3).

After 20 epochs were performed, the training set showed high accuracy in all grades of sacroiliitis (0.9733–0.9967). Sensitivity and specificity also showed high diagnostic performance across all grades (0.9259–0.9744, 0.9727–1.0000, respectively).

The validation set showed a slight decrease in diagnostic performance compared to the training set, but overall showed a very high level of accuracy (0.8984–0.9740). Similarly to the case of the training set, sensitivity and specificity also showed high diagnostic performance regardless of the grade (respectively, 0.6528–1.0000, 0.8879–1.0000, respectively). Referring to the Grad-CAM image, we can see that overall, the deep-learning model was able to locate and grade the sacroiliac joints despite undergoing unsupervised learning. Positive samples are shown in Figure 4.

Both positive and negative predictive values (PPVs and NPVs) exhibited high figures across the datasets: for the training set, the PPV ranged from 0.8621 to 0.9744 and the NPV ranged from 0.9826 to 0.9962. In the validation set, the PPV ranged from 0.7800 to 1.000, and the NPV ranged from 0.8915 to 1.0000. Detailed figures can be found in Table 4.

Diagnostic accuracy was high across the board for both the training and validation sets, regardless of the grade (Figure 5).

## 4. Discussion

Low back pain or hip pain is prevalent among adults, with the majority stemming from mechanical causes. In diagnosing axSpA, it is crucial to differentiate between inflammatory and mechanical back pain. The understanding of primary healthcare providers regarding spondyloarthritis plays a pivotal role in detecting the disease early. Research indicates that directing patients with inflammatory back pain to rheumatologists facilitates prompt diagnosis. However, the misuse of certain imaging techniques has contributed to delays in diagnosis. The significance of patient–physician referral in this context has been underscored in various studies [21].

Although magnetic resonance imaging and computed tomography are used to detect sacroiliitis in developed nations, plain radiograph images still hold a significant position. In many countries, plain radiographs are the primary and often sole imaging technique for evaluating patients with axSpA. This is because MRI is costly and not universally accessible. However, it is widely acknowledged that conventional radiographs are not consistently dependable for detecting sacroiliitis [22]. In this study, we used a large and unique dataset to train, validate, and test the model. The resulting performance was at least as good as (but most likely better than) the performance of experienced readers. The convolutional neural network was able to achieve nearly the same level of performance on both the validation and training sets, indicating a high level of diagnostic performance of the model. Hence, our model serves as a supplementary diagnostic tool in clinical settings and has demonstrated potential as a classification instrument in research endeavors centered on axSpA patients.

The use of AI in diagnosing sacroiliitis in plain radiographs presents several advantages. First, AI algorithms, particularly deep-learning models, can analyze large amounts of data and identify patterns that may not be readily apparent to human observers. This ability to process and learn from vast datasets allows AI to potentially outperform human experts in diagnostic accuracy. Second, AI can provide consistent results unaffected by factors such as fatigue or subjective bias that can influence human interpretation. Third, AI can significantly reduce the time required for diagnosis, thus increasing the efficiency of healthcare delivery. Lastly, AI can serve as a valuable decision-support tool for clinicians, helping them make more informed decisions based on objective data analysis.

In past research, several diagnostic models have been developed across various medical domains, leveraging traditional AI algorithms. Machine learning, a subset of artificial intelligence, excels in identifying patterns from vast datasets [23,24]. This approach has found extensive applications in medical imaging, enabling classification tasks ranging from melanoma detection [25] and discerning smoking statuses using MRI [26] to diagnosing dermatological ulcers [27], breast cancer [28], lung diseases [29,30], and spotting vertebral compression fractures [31,32]. While computer-assisted analyses can adopt various methods like statistical techniques, instance-based evaluations, decision trees, and artificial neural networks (ANNs), machine-learning models are not without their challenges. Notably, they can be prone to biases in imbalanced datasets and might overfit when confronted with high-dimensional feature vectors. As such, it is vital to tailor and assess machine-learning methods meticulously, ensuring their effectiveness for specific clinical applications

Recently, Bressem et al. developed and tested an AI diagnostic model for the detection of radiographic sacroiliitis using conventional radiographs [15]. They utilized a substantial and distinct dataset for the training, validation, and testing of our model. They utilized the ResNet-50 convolutional neural network architecture as a base model, pre-trained on the expansive ImageNet-1k dataset comprising more than 1.28 million images. The outcomes showed that their model’s efficiency was on par with, if not superior to, that of a skilled professional in radiographic sacroiliitis evaluation. This neural network demonstrated a consistently high performance across both training and validation datasets, underlining its reliability and sturdiness. Consequently, with this model, they achieved an excellent model accuracy on the validation data.

But, many of these studies in the field utilize intricate data collection methods and require knowledge of sophisticated coding tools. This complexity poses challenges for many medical researchers unfamiliar with the nuances of diverse coding systems and advanced learning platforms. Instead of using conventional coding tools, we utilized a simple no-coding platform named DEEP:PHI. DEEPNOID, a Korean medical AI startup, has developed this research platform which is currently available online (https://www.deepphi.ai, accessed on 11 December 2023). Many doctors are leveraging this platform to carry out artificial intelligence research. DEEP:PHI features a user-friendly GUI, enabling users to conduct image preprocessing, generate neural network models, and verify training results—all within its workflow window. Furthermore, the platform’s server offers a robust research environment that does not rely on local GPUs or hard disk drives. Being web-based, this platform facilitates collaboration among doctors and developers from different organizations, allowing them to gather data and conduct joint research online. The platform allows for the effortless uploading of research models, catering to a wide range of AI research requirements. Should there be a need for customization, users can modify or craft modules specific to their studies using the built-in code editor. Being freely accessible online, users can seamlessly upload labeled data and then employ an intuitive palette to connect functions such as image processing, training, and validation, allowing them to customize the learning process to fit their objectives. Previously, Lee et al. studied performance evaluation in [^18^F] Florbetaben brain PET images’ classification based on this platform and showed promising results [33].

Meanwhile, in our study, we developed and tested an AI model for the detection of radiographic sacroiliitis on conventional radiographs using this simple no-coding platform. This demonstrated that both the training set and validation set exhibited excellent accuracy across all grades of sacroiliitis, with sensitivity and specificity also demonstrating strong diagnostic performances. Furthermore, we demonstrated the feasibility of our model on a test dataset of novel data, achieving a performance at least comparable to that of two human experts.

This study stands out as a pioneering work in utilizing deep learning to diagnose structural bone change in individuals with axSpA, just by simply utilizing an open web-based AI platform. In the medical realm, applying methods from other domains is not straightforward due to distinct differences based on the organ or disease in question. Our work introduces a deep-learning framework, tailored through transfer learning, capable of effectively diagnosing sacroiliitis with a limited set of simple plain radiographic images. Unlike traditional techniques that necessitate complex repeated preprocessing and coding systems, our approach allows for analysis using only readily acquired images.

Our developing an AI model for diagnosing sacroiliitis has several limitations. One primary concern is the variability in image quality and patient positioning, which can affect the AI model’s accuracy [34]. To reduce this possible confounding factor in the training process of deep-learning algorithms, our study was designed to train the AI model by selecting only the right side of a plain photograph of a sacroiliac joint, making the diagnosis somewhat simpler. We might inadvertently have missed out on capturing the full spectrum of sacroiliitis presentations.

Secondly, the subtlety of early sacroiliitis changes on plain radiographs can be easily overlooked, even by trained radiologists [35], making it challenging for AI models to detect them consistently. We took measures to ensure that our images were interpreted by seasoned radiology staff, but the inherent subjectivity in human interpretation cannot be entirely ruled out. Some studies examined the factors contributing to discrepancies among readers when assessing AP pelvic X-rays based on the modified NY criteria lesion types. The findings demonstrated that sclerosis and erosions exhibited the most significant inconsistencies among readers. Of all the sacroiliitis imaging types, erosions showed the least agreement at 25%, and carried a high odds ratio for divergence in interpretation [36]. To minimize inter-observer and intra-observer variability, we synchronized the X-ray and CT case correlations through a consensus reading by two radiologists. However, early stage sacroiliitis with subtle structural changes may still go unnoticed. This variability could influence the ‘ground truth’ labels, which in turn might affect the training and validation of the AI model. Previous research has attempted to develop diagnostic models for detecting bone marrow edema in the sacroiliac joint, aiming to identify early stage sacroiliitis using traditional deep-learning algorithms, and has demonstrated encouraging results [37]. Future work may involve the development of a diagnostic tool based on MR scans using a no-code platform.

Thirdly, we employed DensNET121, an unsupervised learning approach. Although this selection appears pioneering, it introduces challenges related to interpretability. Unsupervised learning in AI offers the potential to discover hidden patterns in data without predefined labels. However, the downside is a lack of clear interpretability, as AI models can cluster or categorize data in ways that are not intuitively meaningful to humans. In real practice, our model was often correct in its diagnosis when referring to the Grad-CAM images but incorrect in the basis of its judgment or even incorrect in the diagnosis itself (Figure 6). As a result, without the guidance of labeled images, the model may miss important distinctions or make incorrect associations. Unsupervised learning typically demands a vast and diverse training dataset to attain accuracy comparable to supervised learning techniques. In our study, the dataset’s size, encompassing 492 patients, posed a significant limitation. Furthermore, the unsupervised nature of the model means it might establish associations that, while statistically significant, might not always be clinically meaningful. This could lead to potential misinterpretations or oversights in detecting crucial distinctions in the disease’s manifestations.

Lastly, the wide-ranging consequences of incorporating AI into clinical settings must be acknowledged. AI models are often described as “black boxes”, with their decision-making processes being somewhat opaque. This lack of transparency can result in trust issues among both clinicians and patients. An over-reliance on such AI models might result in reduced human oversight, potentially overlooking uncommon presentations or infrequent complications [38]. AI, in its current form, cannot supplant the holistic clinical judgment of healthcare practitioners, given its inability to encompass a patient’s complete clinical context. Thus, integrating AI into clinical operations presents challenges, particularly in terms of regulatory, ethical, and data privacy considerations.

While our model has shown promise within the controlled environment of this study, its performance and applicability in diverse real-world settings remain to be seen. The challenges of diverse patient populations, varying image quality, and the comprehensive clinical judgment that healthcare professionals bring to the table are aspects that AI, in its current form, might not fully encapsulate.

In conclusion, our study emphasizes the potential of AI in diagnosing sacroiliitis using plain radiographs through a no-code deep-learning platform. This approach can serve as a Computer-Aided Diagnosis system, helping clinicians address the limitations of plain radiographs as diagnostic tools, as demonstrated in prior AI models developed with intricate coding. Nonetheless, it is crucial to approach these findings with caution, acknowledging the limitations and identifying the areas for improvement in future research.

## Figures and Tables

**Figure 1 diagnostics-13-03643-f001:**
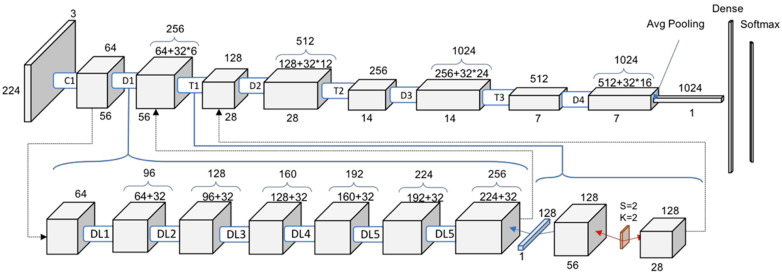
Structure of DensNet 121 with Convolution (C), Dense block (D), Transition blocks (T) and Dense Layers (DL). (symbol * means multiplication sign).

**Figure 2 diagnostics-13-03643-f002:**
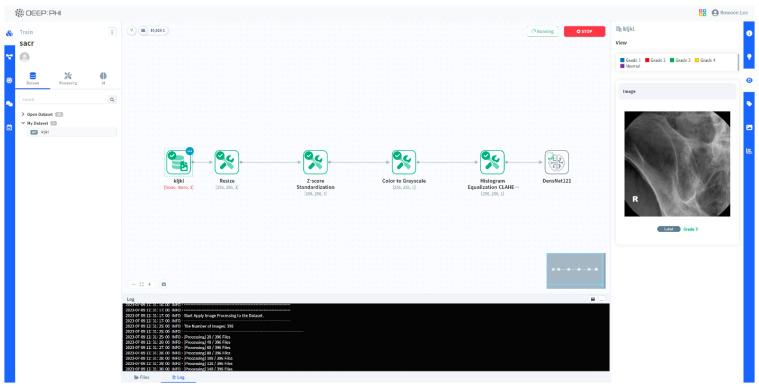
A schematic image of using the DEEP:PHI platform.

**Figure 3 diagnostics-13-03643-f003:**
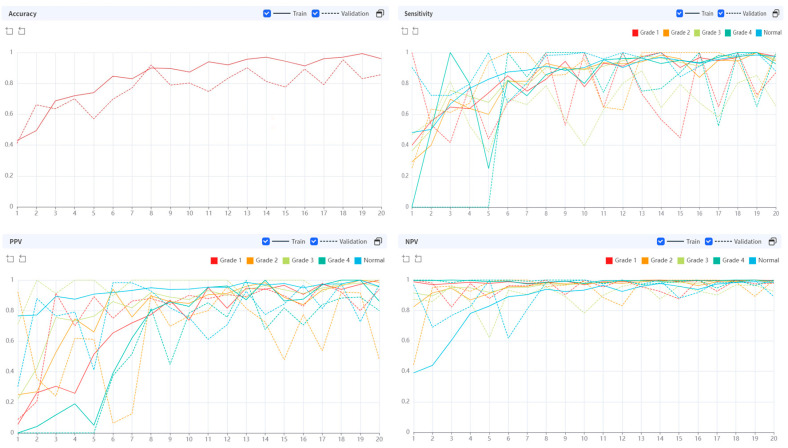
Diagnostic performance of AI algorithm during the training epoch.

**Figure 4 diagnostics-13-03643-f004:**
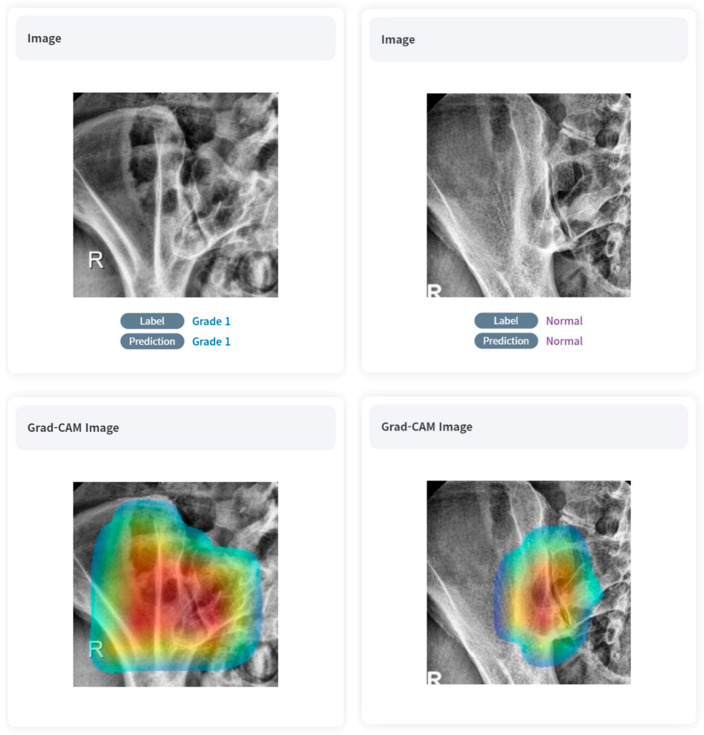
True-positive cases on AI algorithms with Grad-CAM images.

**Figure 5 diagnostics-13-03643-f005:**
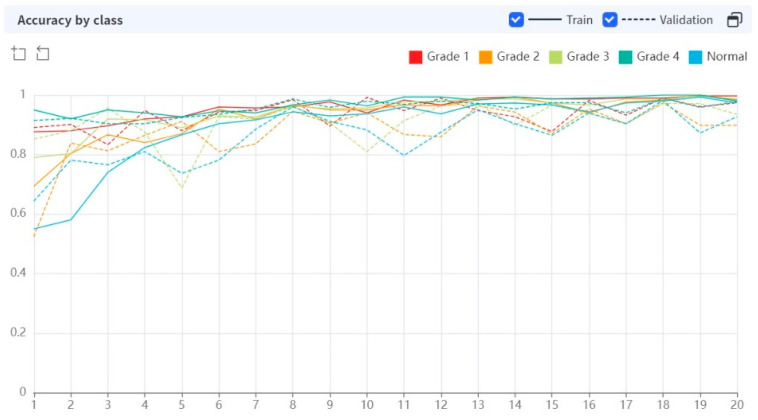
Diagnostic accuracy by class (grade of sacroiliitis).

**Figure 6 diagnostics-13-03643-f006:**
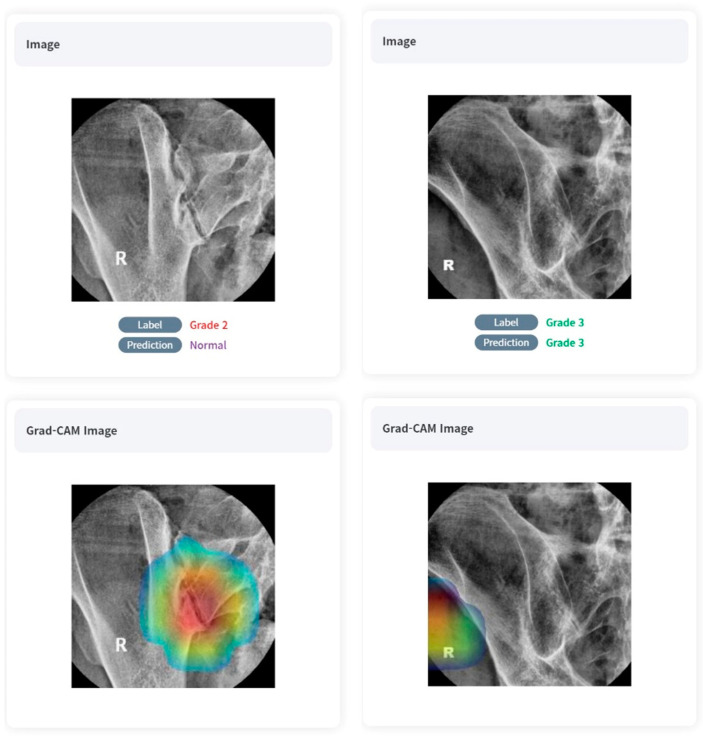
Examples of inappropriate learning and diagnostics of AI algorithm with Grad-CAM images.

**Table 1 diagnostics-13-03643-t001:** Patient demographics of study.

	Non-Sacroiliitis Group	Sacroiliitis Group	*p*-Value
Male	150	122	<0.001
Female	64	156	0.245
Median age	34.2 (range 19.2–42.5)	32.1 (range 18.0–55.2)	0.465

**Table 2 diagnostics-13-03643-t002:** New York criteria for ankylosing spondylitis.

Grade	Changes
0	Normal
1	Suspicious: patchy-like periarticular change
2	Minimal: localized areas with minor erosion or sclerosis, without joint width alteration
3	Definite: moderate to advanced sacroiliitis with evident erosions, sclerosis, joint width alterations (widening, narrowing), or partial ankylosis
4	Severe abnormality: complete ankylosis

**Table 3 diagnostics-13-03643-t003:** Dataset proportion and class ratio.

Proportion			
Training set	Validation set	Test set	Total
300 (61.0%)	96 (19.5%)	96 (19.5%)	492 (100%)
Class Ratio			
Index	Class Name	Count (Ratio)	
0	Grade 1	54 (11.0%)	
1	Grade 2	101 (20.5%)	
2	Grade 3	78 (15.9%)	
3	Grade 4	45 (9.1%)	
4	Normal	214 (43.5%)	

**Table 4 diagnostics-13-03643-t004:** Detailed diagnostic performance of training set and validation set.

	Class	Accuracy	Sensitivity	Specificity	PPV	NPV	F1 Score
Train	Grade 1	0.9967	0.9737	1.0000	1.0000	0.9962	0.9867
	Grade 2	0.9833	0.9452	0.9956	0.9857	0.9826	0.9650
	Grade 3	0.9867	0.9556	0.9922	0.9556	0.9922	0.9556
	Grade 4	0.9800	0.9259	0.9853	0.8621	0.9926	0.8929
	Normal	0.9733	0.9744	0.9727	0.9580	0.9834	0.9661
Validation	Grade 1	0.9740	0.8704	0.9909	0.9400	0.9790	0.9038
	Grade 2	0.8984	1.0000	0.8879	0.7800	1.0000	0.6486
	Grade 3	0.9349	0.6528	1.0000	1.0000	0.9258	0.7899
	Grade 4	0.9792	1.0000	0.9773	0.8000	1.0000	0.8889
	Normal	0.9271	0.8789	0.9742	0.9709	0.8915	0.9227

## Data Availability

The data presented in this study are available on request from the corresponding author. The data are not publicly available due to need approval from the affiliated institution’s DRB (Data review board) is required for disclosure or export.

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
