# Peer review of "The Development and Validation of an AI Diagnostic Model for Sacroiliitis: A Deep-Learning Approach"

_diagnostics, 2023, doi:10.3390/diagnostics13243643_

Round 1
Reviewer 1 Report
Comments and Suggestions for Authors
In this study, authors assessed the AI in the grading of SI joint lesions.
Main problem with the study is the methodology.
Interreader consistency is very low in grading SI joint lesions in the literature. So, it is important how the consensus was achieved in the current study?
Besides, please refer a study that using CT for SI joint grading. As far as I know, there is no CT grading similar to modified New York criteria.
Author Response
Thank you for your insightful comment.
Firstly, two readers interpreted the plane radiographs, CT separately based on the New York criteria. If there was a discrepancy in the grading of sacroiliitis, they would view the images together at same place and discuss to reach an agreement.
The New York criteria is a grading system developed based on plane radiographs. However, there might be false positive or false negative sacroilitis due to degenerative changes, traumatic lesions, or infections. These conditions were excluded by CT(standard reference).
Reviewer 2 Report
Comments and Suggestions for Authors
This paper contains very interesting content.
Especially, the authors describe careful considerations in the discussion paragraph.
However, this paper also contains the following problems.
1.This is a fundamental problem. What kind of disease do the 214 "normal group" patients have? Are they completely healthy volunteer participants?
It is necessary to clarify what kind of group they are.
2. The patient demographics paragraph describes the patients number of each grading, but the author does not mention what the grading is based on. "The New York Criteria" appears in the following paragraphs, but these paragraphs should be re-written to maintain context before and after.
3. As noted in Reference number 37, von Tubergen et al. reported that the gradings of sacroiliits picture were highly variable among evaluators. The author should describe in the method paragraph what the "consensus reading" by the two radiologists that the author wrote on line 310.
Author Response
Thank you for your insightful comment.
1. Patients who visited the rheumatology department for pelvic pain or low back pain but showed no abnormal findings in imaging or blood lab tests, and were not diagnosed with any other diseases, were included in the normal group
2. To ensure consistency in context, incorrectly written numbers, paragraph arrangement, and the order of tables have been corrected.
3. Firstly, two readers interpreted the plane radiographs, CT separately based on the New York criteria. If there was a discrepancy in the grading of sacroiliitis, they would view the images together at same place and discuss to reach an agreement.
Round 2
Reviewer 1 Report
Comments and Suggestions for Authors
My queries were addressed.
Author Response
Thank you for your response. In response to the comments from another reviewer, we have changed the term 'normal group' to 'non-sacroiliitis group'. We hope for a smooth signature process for the review.
Warm regards.
Reviewer 2 Report
Comments and Suggestions for Authors
This manuscript is describing about the developing of an AI diagnostic tool for sacroiliitis. Therefore, what data source the authors used is extremely important.
This reviewer mentioned that the authors needed to clarify what kind of group they described as "normal," but the authors' response has not improved from the previous explanation.
I don't understand what "radiologically and clinically normal" means. Is CT or X-ray imaging performed on“clinically normal”at your hospital? I think this group is a group for whom CT and X-rays were taken for some clinical abnormality, but no abnormality was detected in the images. Am I wrong? Or does this phrase indicates a group of normal volunteers? Or does this phrase indicates "a group of patients diagnosed with SpA but whose presence of sacroiliitis is not visible on plain radiographs or CT"? Of course, if there is no NORMAL in the strict sense in the material, this is a major problem related to the accuracy of this tool, but this can be mentioned as a limitation.
For future reference, I would like to point out that it is important to clearly state in a way that editors and reviewers can understand which parts of the manuscript have been revised or changed. If you have made a major rewrite, you should write a comment saying which line was rewritten. This is an etiquette when submitting a peer-reviewed paper.
Author Response
Thank you for your feedback.
I apologize for not previously mentioning the revisions made in response to another reviewer's comments, including corrections of typos.
The images included in the patient groups were collected from individuals who visited for health check-up and at outpatient clinics, involving X-rays and CT scans. The normal group consists of individuals who were excluded from having inflammatory spondyloarthropathy through history taking, physical examination, and negative blood lab markers like HLA-B27, and who showed no evidence of sacroiliitis in imaging(radiologic "normal" sacroiliiac joint).
In essence, as you pointed out, this includes a broad range of patients, from those who are completely normal, identified during routine health check-ups, to people presenting with traumatic or infectious lesions in other pelvic area.
Therefore, we have revised the terminology to 'non-sacroiliitis group' instead of 'normal group'. However, we have retained the term 'normal' in Tables 2 to 4. This is because, in the context of the New York criteria, it specifically refers to the radiologic normality of the SI joint
Warm regards
Round 3
Reviewer 2 Report
Comments and Suggestions for Authors
The authors treated images of "patients" that have been checked by experts and no abnormalities found as normal images. The task is to have the AI copy the judgment methods used by image experts, but the authors' explanation is only one line (line 122), making it extremely difficult to understand. I think it would be easier to understand if the sentences below line 121 were moved to “Methodology”. Didn't RWL and KHL make the decision to call them "normal"?
Author Response
Q. The authors treated images of "patients" that have been checked by experts and no abnormalities found as normal images. The task is to have the AI copy the judgment methods used by image experts, but the authors' explanation is only one line (line 122), making it extremely difficult to understand. I think it would be easier to understand if the sentences below line 121 were moved to “Methodology”. Didn't RWL and KHL make the decision to call them "normal"?
A: Thank you for the valuable advice from the reviewer. The term for the group without confirmed sacroiliitis has been changed from 'normal' to 'non-sacroiliitis' group. The reason for this change, as previously pointed out by the reviewer, is based on the judgment that it may not be appropriate to categorize 'patients' visiting the hospital as 'normal.' I would like to express my gratitude once again for the your advice on the inappropriate terminology.
And, I have also elaborated on the reasons and methods for categorized this group (non-sacroilitiis) in greater detail. As you suggested, this content has been moved to the methodology section.
Overall, the reviewer's advice has allowed us to clarify the group names and provide more detail on how we categorize them. Thank you for your meaningful advice.
Round 4
Reviewer 2 Report
Comments and Suggestions for Authors
Since the normal group appears before stratification in the “methodology”, I thought that the authors had prepared a separate normal group. I think the new version is easier to understand than the old one.